# Towards Low-Temperature CVD Synthesis and Characterization of Mono- or Few-Layer Molybdenum Disulfide

**DOI:** 10.3390/mi14091758

**Published:** 2023-09-11

**Authors:** Sachin Shendokar, Frederick Aryeetey, Moha Feroz Hossen, Tetyana Ignatova, Shyam Aravamudhan

**Affiliations:** 1Joint School of Nanoscience and Nanoengineering, 2907 E Gate City Blvd, Greensboro, NC 27401, USA; smshendo@aggies.ncat.edu (S.S.); mhossen2@aggies.ncat.edu (M.F.H.); t_ignato@uncg.edu (T.I.); 2Faculty of Nanoengineering, North Carolina Agricultural and Technical State University, Greensboro, NC 27411, USA; faryeete@aggies.ncat.edu; 3Faculty of Nanoscience, University of North Carolina at Greensboro, 1400 Spring Garden St., Greensboro, NC 27412, USA

**Keywords:** chemical vapor deposition, Raman spectroscopy, atomic force microscopy, photoluminescence, scanning electron microscopy, X-ray photoelectron spectroscopy, scanning tunneling electron microscopy, monolayer, MoS_2_

## Abstract

Molybdenum disulfide (MoS_2_) transistors are a promising alternative for the semiconductor industry due to their large on/off current ratio (>10^10^), immunity to short-channel effects, and unique switching characteristics. MoS_2_ has drawn considerable interest due to its intriguing electrical, optical, sensing, and catalytic properties. Monolayer MoS_2_ is a semiconducting material with a direct band gap of ~1.9 eV, which can be tuned. Commercially, the aim of synthesizing a novel material is to grow high-quality samples over a large area and at a low cost. Although chemical vapor deposition (CVD) growth techniques are associated with a low-cost pathway and large-area material growth, a drawback concerns meeting the high crystalline quality required for nanoelectronic and optoelectronic applications. This research presents a lower-temperature CVD for the repeatable synthesis of large-size mono- or few-layer MoS_2_ using the direct vapor phase sulfurization of MoO_3_. The samples grown on Si/SiO_2_ substrates demonstrate a uniform single-crystalline quality in Raman spectroscopy, photoluminescence (PL), scanning electron microscopy (SEM), atomic force microscopy (AFM), X-ray photoelectron spectroscopy (XPS), and scanning transmission electron microscopy. These characterization techniques were targeted to confirm the uniform thickness, stoichiometry, and lattice spacing of the MoS_2_ layers. The MoS_2_ crystals were deposited over the entire surface of the sample substrate. With a detailed discussion of the CVD setup and an explanation of the process parameters that influence nucleation and growth, this work opens a new platform for the repeatable synthesis of highly crystalline mono- or few-layer MoS_2_ suitable for optoelectronic application.

## 1. Introduction

Transition metal dichalcogenides (TMDCs) have a general form of MX_2_ (Metal M = Ti, Zr, Hf, V, Nb, Ta, Re; Chalcogen X = S, Se, Te) and can crystallize into tri-layer 2D structures [1]. The most intriguing TMDC layered element is MoS_2_. Monolayer MoS_2_ has a direct bandgap of 1.9 eV with strong excitonic effects, complementing gapless graphene and exhibiting thickness-dependent properties [2,3,4]. Monolayer MoS_2_ field effect transistors show high current on/off ratios [5]. MoS_2_ attracts attention because it has demonstrated several distinctive optical, transport, and electronic properties. Recent studies show that the thickness-dependent bandgap of MoS_2_ can also be tuned via strain engineering [6,7]. The optical properties of monolayer MoS_2_ are dominated by its excitonic transition characteristics. When excited with solid-state lasers, the photoluminescence (PL) spectrum shows pronounced emission peaks. These luminescence peaks are direct excitonic transitions. No such phenomenon was observed in the bulk form of MoS_2_ [8]. Variations in doping levels can also be employed to engineer the optical properties of MoS_2_. At certain doping levels, the PL spectrum shows the presence of negative trions. A trion is a quasi-state particle consisting of two electrons and one hole [9,10]. To develop applications based on these exquisite properties of MoS_2_, a controlled repeatable synthesis process to produce the optimal crystalline quality is a necessity.

The purest form of monolayer MoS_2_ can be extracted into a monolayer by using exfoliation methods. However, exfoliation methods lack control over size, shape, and thickness uniformity. Large-area and high-quality monolayer MoS_2_ samples can be grown using the CVD method [11,12]. CVD enables many potential applications for monolayer MoS_2_ [13]. Based on the placement of the precursor materials in the furnace, the growth method can be divided into sulfur vapor transport and vapor deposition techniques [14,15]. With sulfur vapor transport, the source materials used are usually Mo, MoO_3_, or (NH_4_)_2_MoS_4_ [16,17]; these are coated on the substrate, which is followed by sulfurization in a high temperature range from 700 to 900 °C in a one- or two-step heating zone unit [18]. These sulfurization methods produce low crystal quality, a small domain size, and poor surface coverage. Thus, the MoS_2_ samples produced are not suitable for a wide range of electronic applications. To harness the potential of monolayer MoS_2_, it is critical to grow high-quality, highly uniform, large-domain-size MoS_2_ over a large area of the substrate [19]. There should be ease in transferring the monolayer MoS_2_ from the growth substrate without degradation in its quality. CVD synthesis is typically carried out at an elevated temperature of around 1000 °C. The high-temperature solid-precursor CVD growth of MoS_2_ is a deterrent for the 3D integration of semiconductor layers, making it a prospective alternative, as the lateral contraction of device features is at a limiting condition [20]. If MoS_2_ logic devices are to be a viable alternative in 3D integration, compatibility between the initial and subsequent layers for deposition requires the low-temperature synthesis of MoS_2_ thin films to reduce thermal stresses and restrict dopant as well as interfacial diffusion [21,22]. With 3D integration, MoS_2_ monolayer Field Effect Transistors (FET’s) can have high electron mobility, enabling faster switching speeds than traditional silicon transistors [23,24]. CVD-grown MoS_2_ monolayers have also demonstrated low-power, energy-efficient logic device capabilities [25]. Due to its mechanical stability, monolayer MoS_2_ is leading the innovation in flexible and wearable electronics, which usually have polymeric substrates requiring low-temperature processing [26]. Thus, this research demonstrates a CVD method for the synthesis of MoS_2_ at a lower temperature of 650 °C. The mono- or few-layer MoS_2_ synthesized during this research demonstrated highly crystalline, stoichiometric qualities, uniform thickness, and large-area growth on Si/SiO_2_ substrates.

## 2. CVD Process Setup

A schematic of the benchtop single-chamber CVD at the Joint School of Nanoscience and Nanoengineering (JSNN) is outlined in Figure 1. The material system and procedure in the subsequent sections have been optimized to provide repeatable monolayer MoS_2_ deposition on a Si/SiO_2_ substrate. The positioning of the sulfur boat, flow rate of Ar, temperature ramp-up rate, and forced cooling of the furnace are critical to ensuring the consistent deposition of MoS_2_ down to monolayers or a few layers.

There is a restricted time zone to avoid the bulk deposition of MoS_2_, achieved by drastically reducing temperature. The material system and control of these process parameters is discussed in detail in the subsequent sections [27,28].

### Material System

MoO_3_ powder—molybdenum (IV) oxide, 99.97%, 25 gms, Aldrich (St. Louis, MO, USA);Sulfur powder—sulfur, 99.98%, 50 gms, Aldrich;Substrate—Si/SiO_2_ (100), 100 mm diameter, P/Boron, 500 µm, from University Wafer (Boston, MA, USA);Seeding promoter—3,4,9,10-perylene-tetracarboxylic acid-dianhydride (PTCDA) [29];Ar gas Supply—Ultra-high purity, AR UHP 300, 336 CF, Airgas (Radnor, PA, USA);Two ceramic boats—0.5 in wide and 2 in long;Weighing scale with the lowest count of 0.01 mg.

To prevent contamination, the ceramic boats are rinsed with acetone, dried with N_2_, and heated in an oven for 5 min at 75 °C. To maintain the stoichiometric ratio of Mo:S, 15 mg of MoO_3_ and 85 mg of sulfur powder are used, as shown in Figure 2. From the physical aspect of conducting an experiment, mass proportion (mg) is mentioned. However, 15 mg of Mo in MoO_3_ with a molar mass of 143.94 g/mole is equivalent to 0.0001 mole of Mo. Similarly, 85 mg of sulfur, which has a molar mass of 32 g/mole, is equivalent to 0.00266 moles. Thus, the molar ratio of these mass proportions would translate to approximately 1:27 Mo:S. A higher mole of sulfur is essential in CVD considering that there is no directed availability of sulfur atom flux at the substrate interface, as the molybdenum trioxide evaporates. The precursors, which are 18 cm apart in a one-inch single-chamber quartz tube, have different evaporation temperatures. The distance of 18 cm ensures that the differential evaporation temperature requirements of sulfur and MoO_3_ are met. This differential positioning of the precursor boats determines the accessibility of the sulfur atoms for the efficient reduction of MoO_3−x_ molecules to MoS_2_. The MoO_3_ powder used as a solid precursor in the conventional CVD has a melting point of 775 °C and a vapor pressure of 10^−4^ torr at 900 °C [30]. Thus, the CVD synthesis of MoS_2_ from solid precursors is traditionally achieved above 750 °C [31]. In this research, the temperature and Ar flow rate were timed to use the sulfur flux to form intermediate compounds such as MoO_3−x_ and MoOS_x_, which promoted the synthesis of MoS_2_ at a lower temperature of 650 °C. Chemical reactions, intermolecular forces, and thermodynamic considerations all play a role in determining how the presence of sulfur affects the evaporation temperature of MoO_3_ [32]. The process’s primary mechanism is the reaction of Mo and S species with the Si/SiO_2_ substrate governed by the controlled transport of sulfur flux, which causes cooling of the intermediate compounds of Mo, resulting in the evaporation of MoO_3_ at a temperature lower than its melting temperature. DSC and TGA analysis can evaluate the chemical kinetics of the process to reason out lowering of the temperature.

## 3. Parametric Aspects of Obtaining Uniformly Large MoS_2_ Crystals Repeatably

There have been several efforts in the past to deposit MoS_2_ through the sulfurization of MoO_3_, and those referred to in this paper are summarized in terms of their process parameters in Table 1. However, the parameters used vary significantly (particularly those highlighted in red); furthermore, there is limited discussion on the parametric interaction except for the novelty that is pertinent to each work. Of particular significance is the distance required between the precursor and deposition time to obtain monolayer MoS_2_, which needs detailing. The parameters for this research, as listed in Table 1, have resulted in the repeatable synthesis of crystalline MoS_2_ monolayers on a large area of 2 cm^2^ (confined due to the size of the tube/boat) at a relatively low temperature of 650 °C. Thus, the novelty of this article lies in its detailed discussion on the process parameters for the repeatable low-temperature synthesis of MoS_2_. This section, in detail, discusses various aspects associated with CVD setup for the tuning of process parameters to achieve the repeatable synthesis of monolayer MoS_2_.

### 3.1. Temperature Control

Thermal energy is needed for evaporating the precursors and heating the substrate. Once the MoO_3_ and S powder is evaporated, the heated substrate provides the energy required for the physisorption of Mo and S molecules. If the activation energy is favorable, the Mo and S molecules will undergo association, dissociation, and ligand exchange to form intermediates of MoO_2_S or MoOS_2,_ finally stabilizing as MoS_2_. Time coincidence, suitable temperature, and ample availability of sulfur flux are essential for MoS_2_ deposition. The control over the energy of physisorption and chemisorption depends on the heating and cooling rate. The heat energy further impacts the crystallinity of the molecules, the uniformity in their thickness, and the diffusion of nuclei into the large MoS_2_ crystals, which coalesce into the monolayer. 

The furnace heating cycle recipe decides the heating and cooling rate for programming the furnace, which coincides with the ramping steps, as shown in Figure 3.

Figure 3 shows the ramping up from room temperature to 650 °C in 25 min, followed by a deposition time of 1–3 min, and then, programmed or accelerated forced cooling. The red line shows the manual open-top accelerated forced cooling procedure. The blue cooling line shows the alternative path for determining the deposition time based on the ramp-up rate, followed by programmed cooling to room temperature. The deposition time rapidly affects the nucleation and growth mechanisms. Considering that the quartz tube CVD is a rudimentary setup, the evaporation of sulfur and Mo, then reaction of those atoms to deposit and form a monolayer MoS_2_, is very sensitive to deposition time, which requires monitoring and adjustment. Thus, the deposition time is specified within a range of 1–3 min (60–180 s), which depends on the rate of heating (15 °C/min to 25 °C/min) and the availability of sulfur flux, which vary in distance (15–18 cm) depending upon the rate of heating.

### 3.2. Position of Sulfur Boat

Positioning the sulfur boat is one of the most critical tasks. The relative position of the sulfur boat with the MoO_3_ boat determines the timely evaporation and availability of sulfur flux to coincide with MoO_3_ evaporation. The sulfur boat heating depends on the conduction, convection, and radiation from the heating coil of the furnace. The Lindberg furnace is programmed for the central part where the MoO_3_ boat is placed. Depending on the maximum temperature and ramp-up rate used for MoO_3_, the position of the sulfur boat is to be adjusted for the sulfur to achieve a vaporization temperature of 200 °C. The physical positioning of the boat away from the center of the furnace is the only way to control the temperature required for the vaporization of sulfur to coincide with the evaporation of MoO_3_. Besides the vaporization temperature requirement of 200 °C for sulfur, the timing of evaporation and the drift velocity of the sulfur atoms are also critical, as sufficient S atoms should be available for the stoichiometric reduction of MoO_3_ to MoS_2_. The defects in MoS_2_ are mostly due to sulfur deficiency, and the defect density for MoS_2_ is 10^14^ per cm^2^. The accurate positioning of the sulfur boat reduces the defects due to efficient S atom flux. The sulfur’s empirically determined precise position is about 15–18 cm upstream of the MoO_3_ boat for a ramp-up rate of 25 °C/min to achieve a final temperature of 650 °C in 25 min. If the final temperature changes or the ramp rate changes, it affects the timing of sulfur vaporization, and thus, the position of the sulfur boat is to be altered. 

### 3.3. Ar Flow Rate

The one-inch quartz tube used for CVD is acetone-cleaned and dried via initial heating to 150 °C. The suction pump creates a vacuum and removes any suspended particulate matter. The Ar environment is maintained at an Ar flow rate of 1000–3000 square cubic centimeters per minute (SCCM). A ceramic cylindrical piece is kept near the inlet, and a dovetail collet end cap is provided to spread Ar flow evenly across the cross-section of the quartz tube. The interruption to the Ar flow inlet provided by the ceramic piece also restricts blowing off of the sulfur powder before reaching its vaporization temperature. The timing and flow rate of Ar affect the mobility and transfer of the sulfur atoms based on the vacuum pressure inside the quartz tube. As the Ar fills the quartz tube, the pressure gauge reading will increase initially. After the quartz tube is filled, the pressure gauge reading stabilizes between 65 and 75 Torr. Once the pressure gauge stabilizes, the Ar flow should be reduced, and it should be just enough to cause the flow of evaporated sulfur atoms to reach the Mo atoms. Thus, once the pressure gauge reading stabilizes and the temperature reaches about 600 °C for MoO_3_, an Ar flow of 100–200 sccm is sufficient to cause the drift of sulfur to Mo. 

### 3.4. Suction Pump

Once the end caps are firmly secured, the suction pump is turned on to remove the suspended particulate matter. The vacuum pump and the heating cycle are switched on simultaneously. The suction pump should run and create a vacuum as the quartz tube temperature reaches 120–150 °C. When the temperature reaches 150 °C, the suction pump should create vacuum pressure inside the quartz tube to the tune of 65–75 Torr. The Ar flow is regulated once the suction pump is off so the pressure gauge displays a pressure of 65–75 Torr. This pressure is conducive to the mobility and transfer of S atoms to the Si/SiO_2_ substrate as Mo atoms, too, are available.

### 3.5. Monitoring and Control

Once the heating cycle is turned on, the temperature is digitally displayed as the furnace is heated. The furnace heating thermostat and display temperature closely follow if the temperature ramp-up is 15 °C/min or less. The pressure inside the quartz tube should be closely monitored as the temperature rises. In solid-precursor quartz tube CVD, the two precursors (S and Mo) will react as they are deposited on the substrate. The physisorption and chemisorption processes involved in the nucleation and growth are a function of the temperature, pressure, and time of the precursor flux availability. The higher ramp rate of 25 °C/min has been empirically determined to obtain large scale monolayer MoS_2_ deposition by causing coincidence of suitable temperature, time and flux of precursor atoms. Following the completion of the heating ramp-up cycle, the temperature for deposition, the flux of S and Mo precursors, which is dependent on the flow rate of Ar, and the cooling rate determine the thickness of the MoS_2_ film deposited. For accelerated cooling, the furnace’s top cover is opened to expose the hot quartz tube to room temperature. This restricts deposition to the monolayer by limiting Mo atoms’ availability due to rapid cooling. Adequate precautions should be taken while unmounting the hot quartz tube and removing the Si/SiO_2_ substrate.

### 3.6. Effect of Sulfur Flux on the Quality of MoS_2_ Deposition

Our objective was to compare the quality of mono- or few-layer crystals of MoS_2_ by growing them in both sulfur-deficient and sulfur-rich atmospheres. For the direct sulfurization of MoO_3_ mono- and a few-layered MoS_2_, the Si/SiO_2_ substrate was pre-treated with PTCDA. While we only used one seeding molecule (PTCDA), variation in the ratio of sulfur precursors was experimented with, yielding highly crystalline, large-size MoS_2_ deposition. Other than the precursor ratio, the control over the position of the boats, Ar flow rate, and ramp-up rate of thermal cycles determine the quality of the MoS_2_ crystals grown. Also, this growth process is a single-step, simple, less costly, and comparatively low-temperature method to obtain repeatable high-quality MoS_2_ crystals. 

## 4. Conformance of Mono- or Few-Layer MoS_2_ with Quality Characteristics

Several characterization techniques were employed to validate the repeatability of the experimental process parameters determined for the growth of MoS_2_ with high crystalline quality. A Carl Zeiss Auriga BU FIB FESEM detector was used to identify and image the monolayer and few-layer MoS_2_. The in-lens detector of the FIBSEM delineates the morphological differences between the monolayer and few-layer MoS_2,_ as shown in Figure 4. 

One of the primary factors for determining the quality of growth is the Mo:S atomic ratio. The optimum mass ratio between Mo and S atoms for repeatable growth experimentation was empirically derived to be nearly 1:6. Thus, 15 mg of MoO_3_ and 85 mg of S powder have been adopted as the norm for the high-quality stoichiometric low-temperature CVD deposition of MoS_2_. The Mo:S mass ratio of 1:6 is equivalent to molar ration of 1:26. A significantly higher amount of sulfur is needed to fill up the quartz tube and ensure optimum sulfur flux availability.

While evaluating the nucleation and growth processes, the SEM micrographs identify that MoS_2_ domain growth starts from a hexagonal nucleus with three sides of Mo and S terminations each. S terminations grow comparatively faster in a Mo-rich or S-deficient atmosphere as they are more energetically unstable, resulting in a triangular domain with Mo terminations. For the second scenario, when the Mo:S ratio corresponds to the stoichiometric ratio of MoS_2_, both growth rates are equal, and the final shape is a hexagon. Finally, for the greater-than-1:2 ratio, we also have a triangular domain, but with S terminations. It has been reported that triangles with Mo terminations have sharper, straighter edges than S triangles [33]. Figure 4 demonstrates the large sizes of the triangular MoS_2_ grains in (a) with the increase in the density and coalescence of the triangular grains in (b) and (c). Finally, the few-layer MoS_2_ grains with overlapped triangles are shown in (d). 

A alpha300R Confocal Raman Microscope (WITec GmbH, Ulm, Germany), objective Zeiss EC Epiplan-Neofluar 100×/0.9 equipped with a 532 nm laser and two gratings, 1800 g/mm for Raman and 300 g/mm for photoluminescence (PL), was used for the structural characterization of MoS_2_ flakes.

Raman spectroscopy is widely used to determine the quality of MoS_2_ samples [34,36,37,38,39,40,41,42]. MoS_2_ exhibits two characteristic Raman bands at around 402 cm^−1^ and 383 cm^−1^: out-of-plane (A_1g_) and in-plane (E_2g_) vibrations. The separation between these peaks can be used to identify the layer number: 19–20 cm^−1^ corresponds to a monolayer (Figure 5c, red curve below), and 20–25 cm^−1^ corresponds to an increase from double-layer to bulk MoS_2_ (Figure 5c, blue curve below). During the transition from monolayer to bulk, the out-of-plane Raman mode softens by undergoing a blue shift, while the in-plane Raman mode stiffens by undergoing a redshift [43,44,45].

As depicted in Figure 5c (blue curve), nucleation seeds sometimes form multilayer MoS_2_ with peak separation at 23 cm^−1^. We performed Raman imaging to check the spatial homogeneity of Raman bands over the MoS_2_ flake, defined as a pristine monolayer. The intensity variations in the E_2g_ (383.4 cm^−1^ (Figure 5b)) and A_1g_ (402.5 cm^−1^ (Figure 5a)) peaks were negligible, indicating the uniform thickness of a large area. For accuracy, we ran measurements at a laser power no higher than 2 mW and an acquisition time of 0.1 s to prevent sample overheating.

A photoluminescence map was recorded from the same molybdenum sample area (Figure 6a). It shows significant quenching for multilayer (Figure 6b, blue spectrum) and bulk (Figure 6b, cyan spectrum) MoS_2_, not only due to the gradual transition from the monolayer direct bandgap to the bulk indirect bandgap [46,47,48], but also due to defects and terminations [49,50,51]. Monolayer MoS_2_ shows a strong photoluminescence peak at 1.85–1.90 eV (Figure 6b red spectrum). The two peaks observed at 1.8 eV and 1.9 eV are due to direct excitonic transitions at the K-point due to spin-orbit splitting of the valence band [7,50].

Figure 7 shows micrographs acquired using an Asylum MFP-3D Origin, Atomic Force Microscope (AFM), (Oxford Instruments Asylum Research, Santa Barbara, CA, USA) for physical thickness determination of the MoS_2_ monolayers. An AFM tip radius of 2 nm with a resonance frequency of 300 KHz and force constant of 40 N/m was used. Initial survey scans of 20 square microns with 256-point resolution were carried out at 1 Hz. Successive scans were required to locate the edge of the monolayer MoS_2_, after which the scan resolution was increased to 512 points. For the detection of the thickness of the monolayer once the edge was detected, the amplitude retrace was tuned by adjusting the AFM tip drive amplitude for a perfectly overlapped retrace. AFM tip trace path over MoS_2_ crystal is shown by pink line in Figure 7b.

The physical thickness of the mono- and few-layer MoS_2_ was verified based on a 1 nm thickness decrease in the amplitude retrace, as recorded in Figure 7. The sub-nanometric thickness of MoS_2_ confirms that the deposition obtained was a monolayer. Carrying out measurements over the edges of triangular crystals is a tedious process (Figure 7. Pink line). There is a difference between the physical thickness of MoS_2_ monolayer measured by AFM and that determined by Raman assessment, which may be due to the sensitivity of the AFM tip and vibration damping capacity of the AFM setup. 

Surface chemical analysis was performed using the Thermo Fisher ESCALAB 250 Xi (Waltham, MA, USA) X-ray photoelectron spectroscopy setup on supported MoS_2_ samples. We determined the stochiometric ratio of Mo:S from X-ray photoelectron spectroscopy in Figure 8 below. This shows the stoichiometric ratio of Mo:S to be 1:3, less than 1:2, producing a truncated monolayer triangle [19,52]. Other elements present in monolayer MoS_2_ island flakes were oxygen, carbon, and silicon. XPS analysis reveals silicon and oxygen as the background of these MoS_2_ crystals, which is the composition of the silicon oxide layer on top of the silicon. 

The peak at 227.10 is due to a S_2s_ electron, whereas the peaks at 233.30 eV and 230.13 eV coincide with Mo3d_3/2_ and Mo3d_5/2_ of MoS_2_ (seen in Figure 8a. as deconvoluted red and black curves are perfectly overlapped by the XPS response envelope). The peak at 236.46 Mo^+4^ occurs due to MoSi_2_. The S2P_3/2_ and S2P_1/2_ peaks for sulfur (seen in Figure 8b. as deconvoluted as red and green curves respectively) at 162.5 eV and 164.4 eV also confirm stoichiometric MoS_2_ [53]. 

CVD-grown MoS_2_ was transferred from the Si/SiO_2_ substrate using the wet KOH transfer method for TEM analysis. For MoS_2_ transfer onto the TEM grid, samples were first spin-coated at 1500 rpm with PMMA A2, resulting in a 100 nm thick polymer film. These were detached in a 1 M KOH solution, washed several times in deionized (DI) water, and transferred onto TEM grids, and atomic resolution images were acquired using a Nion Utra STEM^TM^ 100, Kirkland, WA, USA setup of suspended molybdenum disulfide samples. For high-resolution TEM (HR-TEM), we used a Quanta foil orthogonal array of 1.2 μm diameter holes with 1.3 µm separation mounted on a 200 M Cu grid [19]. Figure 9a shows an ADF-STEM image of the monolayer MoS_2_. Figure 9b. is Fourier transform identified by hexagonal lattice and region (purple left top–inset) zoomed in for evaluating the lattice spacing as in Figure 9c. The brighter spots correspond to Mo atoms, and the darker areas correspond to sulfur atoms. In the absence of contamination, the contrast in the STEM mode under the Z contrast relationship allows us to distinguish between atoms with different Z numbers. The Fourier transform images demonstrate the high crystallinity of the MoS_2_ grown. The lattice spacing computed using the standard procedure is about 0.65 nm, which is close to the value published in the past [35].

## 5. Conclusions

Highly crystalline monolayer and few-layer MoS_2_ samples can be controlled va CVD using MoO_3_ and a sulfur precursor at reasonably low temperatures. At a 650 °C growth temperature, there is uniform surface coverage with monolayer island flakes within a short growth duration. The structural characterization of MoS_2_ using SEM, Raman spectroscopy, PL, atomic force microscopy, XPS, and STEM confirmed that the layers are indeed high-quality uniform monolayers and a few layers of molybdenum disulfide synthesized via CVD.

## Figures and Tables

**Figure 1 micromachines-14-01758-f001:**
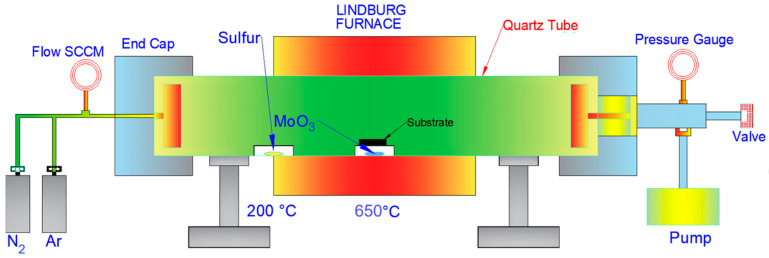
Schematic of CVD Setup.

**Figure 2 micromachines-14-01758-f002:**
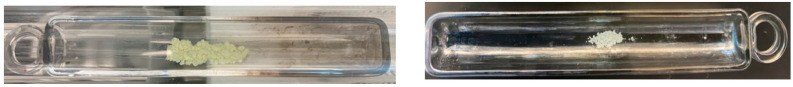
Ceramic boats with sulfur (85 mg) and MoO_3_ (15 mg).

**Figure 3 micromachines-14-01758-f003:**
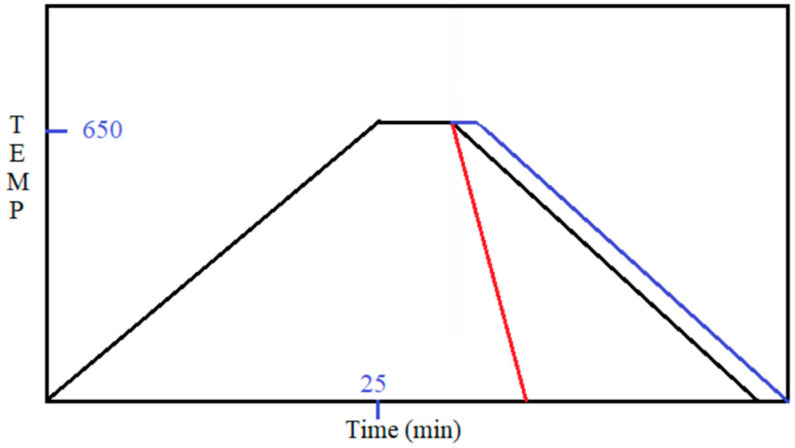
Temperature cycle demonstrating the importance of cooling rate.

**Figure 4 micromachines-14-01758-f004:**
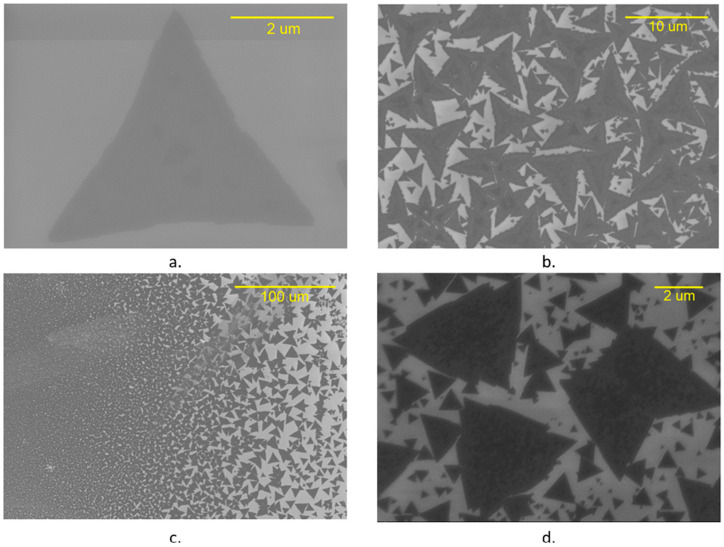
SEM micrographs of MoS_2_, on Si/SiO_2_ substrates (Carl Zeiss Auriga—BU FIB FESEM). (**a**) SEM monolayer MoS2, (**b**) shapes and bi-layer MoS2, (**c**) coalescence of MoS_2_ crystals, (**d**) few-layer MoS_2_ crystals.

**Figure 5 micromachines-14-01758-f005:**
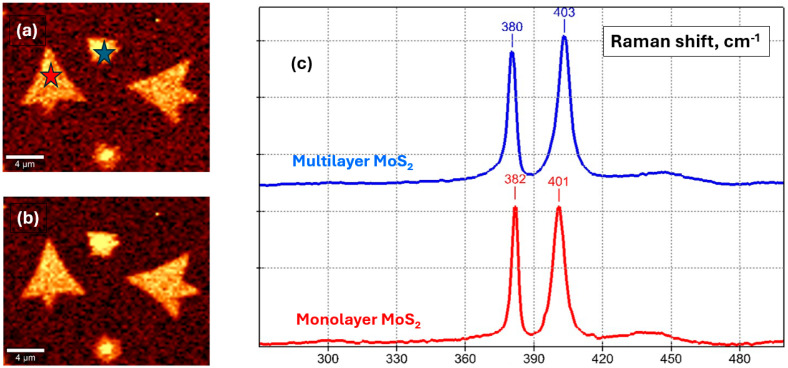
Structural characterization of MoS_2_ on silicon/silicon dioxide. Raman mapping: (**a**) area under E_2g_ 382 cm^−1^, and (**b**) area under the A_1g_ 403 cm^−1^ peak; (**c**) Raman spectra in the red and blue star areas, respectively (monolayer peaks in red: 19 (1/cm), multilayer peaks in blue: 23 (1/cm)).

**Figure 6 micromachines-14-01758-f006:**
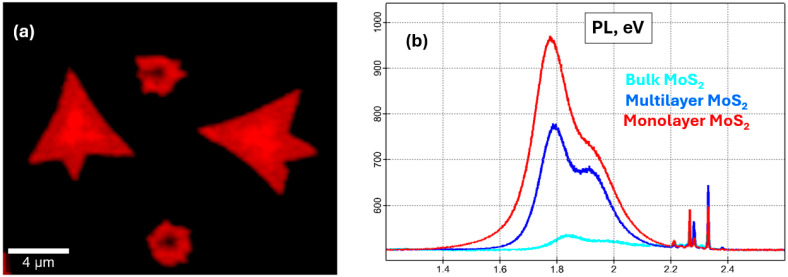
Photoluminescence of monolayer MoS_2_. (**a**) Raman map for MoS_2_ sample area; (**b**) PL spectrum for ML, multilayer, and bulk MoS_2_.

**Figure 7 micromachines-14-01758-f007:**
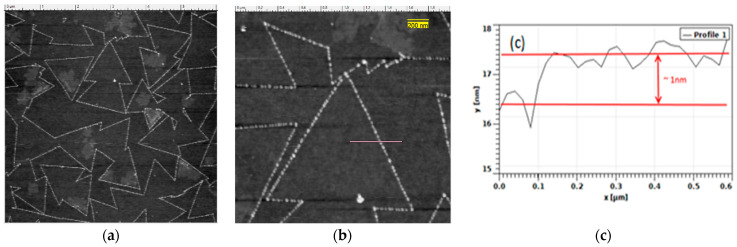
MoS_2_ synthesized on Si/SiO_2_ substrate. (**a**) AFM images of monolayer MoS_2_ flakes; (**b**) AFM images of individual monolayer MoS_2_ flake with AFM tip trace path (pink line); (**c**) thickness measurements of monolayer MoS_2_ along the blue line.

**Figure 8 micromachines-14-01758-f008:**
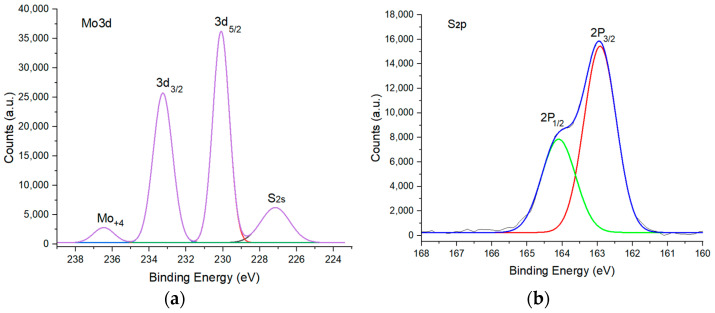
X-ray photoelectron spectroscopy of monolayer MoS_2_. (**a**) Mo3d deconvoluted peaks. (**b**) Sulfur deconvoluted peaks.

**Figure 9 micromachines-14-01758-f009:**
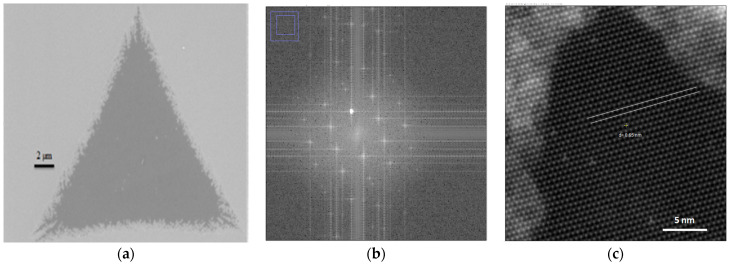
(**a**) ADF-STEM image of monolayer MoS_2_. (**b**) Fourier transform image of MoS_2_ lattice showing high crystallinity. (**c**) Computed lattice spacing.

**Table 1 micromachines-14-01758-t001:** Process parameters for CVD of MoS_2_ using MoO_3_ and S solid precursor.

Precursor-1	Precursor-2	Carrier	Temp	Dist P1–P2	Substrate	Time	Dep. Time	Sz of ML	Raman Shft	PL	Ref.
(mg)	(mg)	(sccm)	(°C/m)	(cm)	Treatment	(min)	(s)	(cm^2^)	(cm^−1^)	eV/nm	No
Mo Thin Film	S (NM)	N_2_ (150–200)	750	NM	SiO_2_—Mo TF Used	90	600	1 cm^2^	20.6	NM	[12]
MoO_3_ (3.2)	S (1500)	Ar (100)	700 (15/m)	17.5	SiO_2_—Acetone (Air Plasma)	51	600	20 µm	NM	NM	[14]
Mo Thin Film	S (NM)	Ar+H2 (70)	1000 (15/m)	NM	c-Sapphire	60	1800	2 in dia	22	627 & 623	[15]
MoO_3_ (NM)	S (NM)	Ar (150)	850	NM	SiO_2_—MoO2 TF	180	NM	12 µm	21	NM	[17]
MoO_3_ (10–30)	S (NM)	Ar (150/60)	800 (10/m)	25	SiO_2_	80	600	300 µm	20	NM	[19]
MoO_3_ (2)	S (100)	Ar (22)	560	26	SiO_2_—PTAS	21	1800	60 µm	20.8	NM	[21]
MoO_3_ (18)	S (180)	Ar (5)	650 (15/m)	16	SiO_2_—rGO, PTAS, PTCDA	43	180	5 µm	20	1.83	[29]
MoO_3_ (15)	S (80)	Ar (10–500)	760 (15/m)	18	SiO_2_—O_2_ Plasma	51	1800	50 µm	20.3	625 & 675	[33]
MoO_3_ (NM)	S (NM)	Ar (130)	800 (25/m)	22	SiO_2_—Acet/2 Propanol	32	1200	1 µm	19.8	625 & 675	[34]
MoO_3_ (100)	S (200)	Ar (100)	850 (13.5/m)	30	SiO_2_—O_2_ Plasma	63	1200	2 cm^2^	20	1.89	[35]
MoO_3_ (15)	S (85)	Ar (100–200)	650 (25/m)	15–18	Si/SiO_2_—PTCDA	25	60–180	2 cm^2^	19	1.8–1.9	TP

NM: Not Mentioned; TP: This Paper.

## Data Availability

The data presented in this study are available on request from the corresponding author. The data is not publicly available due to academic restrictions.

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
