# Peer review of "Towards Low-Temperature CVD Synthesis and Characterization of Mono- or Few-Layer Molybdenum Disulfide"

_micromachines, 2023, doi:10.3390/mi14091758_

Round 1
Reviewer 1 Report
Referee report
The article “Towards low-Temperature CVT Synthesis and Characterization of Mono or Few-Layer Molybdenum Disulfide” is devoted to development of methods for the synthesis of quasi-two-dimensional materials, in particular MoS2. It is quite actual and interesting topic. The advantage of the work is that the authors proposed a synthesis method at a relatively low temperature. They used an adequate set of techniques to characterize the samples and showed good quality of MoS2. The article contains new experimental data and will be interesting for researchers and technologists. The article can be published after minor revision.
Comments:
1) For the stoichiometry of the obtained samples, the molar ratio of chemical elements and chemical compounds is important, but the authors give the mass ratio. The molar ratio must be given. So, the molar mass of molybdenum is 0.09594 kilograms, mass of MoO3 0.14394 kilograms, and the molar mass of sulfur is 0.03207 kilograms. Thus, 15 milligrams of MoO3 contains approximately 0.0001 mole of molybdenum. 25 milligrams of sulfur contains approximately 0.00078 moles of sulfur. That is, the molar ratio of molybdenum to sulfur in this case is approximately 1:8, which is less than 1:2. In addition, in Figure 2, the authors claim that the boat contained 85 milligrams of sulfur, and in the text they write 90 milligrams. So, in the text of the article, the authors should give the molar ratio of molybdenum to sulfur, and explain why such molar ratio was required for the synthesis (Mo:S is 1:8 and less).
2) In terms of the relevance of the work, the authors should refer to a wider range of methods for obtaining quasi-two-dimensional materials. This is not only CVD and CVT, but also atomic layer deposition, radiation methods (see for example – Dvurechenskii et al. Electron–Beam Radiation Effects in the Multilayer Structures Grown by Periodical Deposition of Si and CaF2 on Si (111). Mater. Proc. 2023, 14, 68 https://doi.org/10.3390/IOCN2023-14481) and other methods. Authors should articulate the benefits of their method more clearly.
3) Page 3. What authors mean in “pressure of 10−4 at 900 °C”? Was it Pascals or Torres?
4) The authors are certainly not the first in the world to observe Raman scattering peaks from MoS2. They should provide references to pioneering work in this area.
5) The authors should describe in more detail the conditions for recording Raman spectra. It is known that when laser radiation is concentrated in a spot with an area of several square micrometers, local heating of the sample can occur. It is known that heating leads to a shift of the phonon frequencies towards lower frequencies, due to anharmonism. The possible local heating needs to be clarified.
6) Authors should describe the atomic force microscopy setup and scanning conditions.
7) The article is written quite well and clearly, but authors should clarify all abbreviations used, even if they are common knowledge - for example FET, PL, SEM, AFM, XPS, SCFH, PTCDA and others.
Accept with minor revision.
Author Response
We appreciate the efforts of the Reviewer to analyze the Article text critically. We have rigorously worked on the comments. Raman and PL characterization has been updated with the Expertise of Senior Authors. STEM results have been updated. The reference list has been changed.
The updated article text is maintained in RED color font. The reference list which was 40 earlier is now 56.
The specific response to each comment of the reviewer is attached herewith.
We are thankful to the Reviewers and the Team at MDPI as it seams the Article Quality has been improved for the benefit of the readers. The MDPI process has also been a learning lesson for the Authors.

Reviewer 2 Report
In this article, the authors presented the chemical vapor deposition growth of MoS2. In the last decades, many groups study MoS2 through different methods leading including CVT, CVD, PVD and MBE etc. Still, there is a space for this topic if it can be used for large wafers to fully use in real applications. The authors explain the growth method in a very detailed manner, which are quite good for new researcher in the field to understand it. However, this material can be grown easily by CVD compared to other TMDs. My concern is why the authors focus on MoS2 rather than other materials because what I can see in the article is not new to readers and researchers. Can the author point out the explain the novelty of this work? A few technical comments need to be addressed.
1. The authors said CVT, which is confusing because CVT is only can be used for single crystal growth. For example, the authors can see the articles: https://doi.org/10.1039/D2TC04306H, https://doi.org/10.1021/jp512013n, Micromachines 2018, 9(6), 292; please comment on it. What is the difference between such growth and these crystals?
2. How the authors can detect the monolayer from Raman? Can the authors comment on this point? What we can see is that the Raman shift maybe related to electron-phonon interaction and lattice dynamic.
3. PL is mostly related to the excitonic effect, however, it is seen that the intensity enhanced in monolayer in this article, the authors claim that it is from indirect to direct gap happened from bulk to monolayer. There is any direct evidence to see this effect. The valley exciton will have a role in this effect as well.
4. Can the authors provide a full STEM image, because the current image is cropped by the authors, which can’t be clearly understood that it is the STEM image of their MoS2?
5. The figures format is not uniform in this article.
6. They must revise the format of the figures and make it the same. I suggest the authors combine Fig. 5 & 6 in one, Fig. 7 & 8 in one, and Fig. 9 & 10 in one figure.
7. Can the author estimate the lattice space from the STEM?
Need some improvements
Author Response
We appreciate the efforts of the Reviewer to analyze the Article text critically. We have rigorously worked on the comments. Raman and PL characterization has been updated with the Expertise of Senior Authors. STEM results have been updated. The reference list has been changed.
The updated article text is maintained in the RED color font. The reference list, which was 40 earlier, is now 56.
The specific response to each comment of the reviewer is attached herewith.
We are thankful to the Reviewers and the Team at MDPI, as it seems the Article Quality has been improved for the benefit of the readers. The MDPI process has also been a learning lesson for the Authors.

Round 2
Reviewer 2 Report
1. My first concern about the quality of the article is doesn't considered well, I suggest the authors address the relevant articles and make a suitable comparison that this work is important. What they answered is that all the other groups did better than this. The novelty of this work should be considered as what is new in this article.
2. Second the method they are using is CVD rather than CVT: CVT mostly use some transport agent that can transport the vapour. Thus I suggest the authors reference these articles and write them properly.
3. Comment 2 should be considered by AFM.
4. The new PL shows some additional peaks, the authors need to comment on it.
5. Figures numbers are not updated.
No
